# Field emission effect in triboelectric nanogenerators

Di Liu [1,2,7], Yikui Gao [1,3,7], Wenyan Qiao[1,3], Linglin Zhou [1,3,4], Lixia He[1,3], Cuiying Ye[1,3], Bingzhe Jin[1,3], Baofeng Zhang[5], Zhong Lin Wang [1,3,6] ✉ & Jie Wang [1,3,4] ✉

Triboelectric nanogenerators (TENGs) have garnered increasing attention due to their exceptional ability to convert mechanical energy into electricity. Previous understanding is that the electric performance of TENGs is primarily restricted by contact electrification, air breakdown, and dielectric breakdown effects. Here, we have discovered the occurrence of field emission arising from contact electrification and identified its limitation on surface charge density, subsequently impacting the output performance of TENGs. More importantly, we reveal that field emission occurs prior to dielectric breakdown, introducing a new limitation for TENGs performance. By suppressing the field emission effect, an ultrahigh charge density in contact electrification, reaching up to 2.816 mC m$^{-2}$, is achieved, significantly exceeding previous reports. Additionally, we show that by regulating the field emission effect, TENGs could produce an energy density over 10 J m$^{-2}$. These findings are crucial for improving TENG's performance and enhancing the understanding of contact electrification.

Contact electrification refers to the charge transfer that occurs when two materials come into contact, and one material usually charges positively while leaving opposite charges on the other material. Electrostatic charges generated by contact electrification establish a large electric field on surfaces, conferring remarkable capabilities in diverse applications including electrostatic adsorption and separation, electrostatic printing, electrocatalysis[1–3]. To enhance performance in these applications, a large electrostatic force or a strong electric field is often required, both of which are clearly dependent on a high surface charge density. Recent efforts to generate electricity from electrostatic charges via triboelectric nanogenerators (TENGs) have pushed the search for a high surface charge density to a new level of urgency, as the electric performance of TENGs is also reliant on a high surface charge density[4–6].

Previously, it is widely accepted that the surface charge density of TENGs is determined by the charge generation process due to contact electrification ($\sigma_{\text{contact electrification}}$) and by the charge dissipation process, in which surface charges could be released through air breakdown ($\sigma_{\text{air breakdown}}$) and dielectric breakdown ($\sigma_{\text{dielectric breakdown}}$)[7]. As a result, the output charge density of TENGs

$$\sigma_{\text{TENG}} = (\sigma_{\text{contact electrification}}, \sigma_{\text{dielectric breakdown}}, \sigma_{\text{air breakdown}}) \min \quad (1)$$

depends on the minimum of these three factors, which is a classic phenomenon of "cask effect". In this context, several approaches to enhance the charge density of TENGs have been proposed. Suppressing air breakdown, including the use of ultrathin dielectric materials[8,9], interface lubrication[10,11], and structural design[12,13], can

[1]Beijing Key Laboratory of Micro-Nano Energy and Sensor, Center for High-Entropy Energy and Systems, Beijing Institute of Nanoenergy and Nanosystems, Chinese Academy of Sciences, Beijing 101400, P. R. China. [2]Department of Mechanical Engineering, The Hong Kong Polytechnic University, Hong Kong 999077, P. R. China. [3]College of Nanoscience and Technology, University of Chinese Academy of Sciences, Beijing 100049, P. R. China. [4]Guangzhou Institute of Blue Energy, Knowledge City, Huangpu District, Guangzhou 510555, P. R. China. [5]Hubei Key Laboratory of Automotive Power Train and Electronic Control, School of Automotive Engineering, Hubei University of Automotive Technology, Shiyan 442002, P. R. China. [6]Yonsei Frontier Lab, Yonsei University, Seoul, Republic of Korea. [7]These authors contributed equally: Di Liu, Yikui Gao. ✉e-mail: zhong.wang@mse.gatech.edu; wangjie@binn.cas.cn

significantly increase the surface charge density, even by orders of magnitudes[6]. These findings suggest that air breakdown often determines $\sigma_{TENG}$. Furthermore, by fully eliminating air breakdown under vacuum conditions, the surface charge density can reach values exceeding 1 mC m$^{-2}$ [5,7]. Additionally, the charge pump technology has been introduced to overcome the limitations of contact electrification, leading to a high effective charge density of up to several mC m$^{-2}$ [14–16]. In accepted studies, without the constraints of contact electrification and air breakdown, dielectric breakdown is considered as the next ceiling that determines the maximum surface charge density in TENGs, due to low dielectric properties of common polymers[7,17–20]. Are there other physical mechanisms that determine the charge density limit of TENGs as the surface charge density continues to increase? The answer could provide insights for designs of high-performance TENGs.

Here, we have discovered the field emission effect resulting from contact electrification and demonstrate its limitation on the maximized charge density of TENGs under vacuum conditions. We obtain an ultrahigh charge density from contact electrification up to 2.816 mC m$^{-2}$ via suppressing field emission, which significantly exceeds previous reports. On the other hand, we confirm the ubiquity of field emission in TENGs. More importantly, we reveal the occurrence of field emission prior to dielectric breakdown in TENGs, particularly in dielectric materials exhibiting higher dielectric breakdown strength, and provide a modified charge density governing equation for TENGs. These findings will enhance the understanding of the upper limit of charge density in contact electrification and guide the optimization of TENGs' performance, which is also expected to arouse broad interests in diverse scientific communities and application fields relating to the research of contact electrification.

## Results

### Hypothesis of field emission in TENGs

The working mechanism of TENGs is based on the coupling of contact electrification and electrostatic induction for converting periodic mechanical motions into alternating electric signals in the external circuit (Fig. 1a). Based on this principle, TENGs can also provide a simple, in-situ, and real-time technique to probe charge transfer during contact electrification. Ideally, the output charges of a TENG, also referred to as the transferred charges in the external circuit, are equal to the quantity of triboelectric charges if there are no charge dissipations (Fig. 1b). However, surface charge dissipation due to air breakdown has been widely observed in TENGs, resulting in output charges of TENGs are often smaller than triboelectric charges, corresponding to the residual surface charges[5,6,11,21–23]. By suppressing air breakdown, as described by Townsend avalanche, the maximum surface charge density in TENGs can be elevated to surpass 1 mC m$^{-2}$ [5].

It is noted that during repeated contact and separation, the charge accumulation and dissipation processes of the dielectric film reach a balance, resulting in the stable output charge of TENGs, as depicted in Fig. 1c. Then, we extend the contact time of two triboelectric layers to increase the triboelectric charges as much as possible, resulting in a high output charge density of 2.816 mC m$^{-2}$ in vacuum (Fig. 1d, f and Supplementary Fig. 1)[5,7,8,12,13,24–27]. As shown in Fig. 1e, after separation, the high output charge density quickly decreases because of the charge dissipation for the dielectric film (Supplementary Fig. 2), and then charge accumulation starts again. Based on these results, we propose the following sequence of events in TENGs: as the two materials contact and their electron clouds overlap strongly, electrons transfer between them (Fig. 1g); during the separating process, charge dissipations such as leakage current and dielectric breakdown, would occur, leading to a decline in surface charge density (Fig. 1h)[5]. In other words, surface charges are in dynamic equilibrium between charge generation, charge dissipation and charge output, with charge generation occurring primarily during the contact state and charge dissipation generally dominating in the separate state. Here, the charge

dissipation like air breakdown is eliminated and leakage current in the polyimide film can be ignored during an instantaneous separation, so an ultrahigh charge density from contact electrification is up to 2.816 mC m$^{-2}$ (Fig. 1f). It is noteworthy that there is still a little part of charge loss in Fig. 1d. This phenomenon is not observed in previous literature, in which the output charge density of TENGs is decided by triboelectrification, air breakdown, and dielectric breakdown. It is the occurrence of field emission that releases a little part of charges in vacuum, thereby decreasing the output charges.

In the conventional triboelectric series[2], while polyimide seems less effective at acquiring electrons compared to polytetrafluoroethylene (PTFE), it can achieve substantial charge transfer with sufficient contact time, and its superior breakdown strength and lower leakage current result in a higher charge density than PTFE once field emission is suppressed. This result not only reminds us the importance of regulating charge dissipations to improve output performance of TENGs again, but also provide a new guide for the future selection of triboelectric materials.

### Critical breakdown electric field for field emission in TENGs

To confirm the field emission effect in TENGs, we systematically analyzed the basic working principle of TENGs. A typical TENG comprises one film dielectric and two electrodes, with one electrode periodically contacting and separating from the film dielectric (Fig. 2a). Here, we utilized the contact-separating TENG (CS-TENG) as an example because of its clear and concise physical model, almost idealized uniform electric field, and excellent stability, etc. Clearly, both the electric field across the gas/vacuum dielectric ($E_g$) and the electric field across the film dielectric ($E_D$) vary with the gap distance. The simplest model of a TENG contains two planer layers: a gas dielectric layer with thickness of $d_g$ and permittivity of $\varepsilon_g$; a film dielectric layer with thickness of $d_D$ and permittivity of $\varepsilon_D$ (Fig. 2b). The surface charge density (SCD) on dielectric material is denoted as $\sigma_D$. In ideal conditions, during the varying gap distance with the transferred charge density in the external circuit of $\sigma_T$, $E_g$ and $E_D$ respectively satisfy the following equations:

$$E_g = \frac{(\sigma_D - \sigma_T)}{\varepsilon_g} \tag{2}$$

$$E_D = \frac{\sigma_T}{\varepsilon_D} \tag{3}$$

Voltage across the gap dielectric ($V_g$) and voltage across the film dielectric ($V_D$) can be respectively described as:

$$V_g = \frac{(\sigma_D - \sigma_T)d_g}{\varepsilon_g} \tag{4}$$

$$V_D = \frac{\sigma_T d_D}{\varepsilon_D} \tag{5}$$

Accordingly, the surface energy density stored in the gas dielectric ($U_g$) and the film dielectric ($U_D$) are described as follows:

$$U_g = \frac{(\sigma_D - \sigma_T)^2 d_g}{2\varepsilon_g} = \frac{\varepsilon_g E_g^2 d_g}{2} \tag{6}$$

$$U_D = \frac{\sigma_T^2 d_D}{2\varepsilon_D} = \frac{\varepsilon_D E_D^2 d_D}{2} \tag{7}$$

Different from the fixed gap distance and the applied voltage source in conventional breakdown studies, the two triboelectric layers are charged at the contact state and then gradually separated with the gap distance varying from nanometer scale to centimeter scale, and the strong electric field is generated by contact electrification rather than the external power source. Therefore, the breakdown process

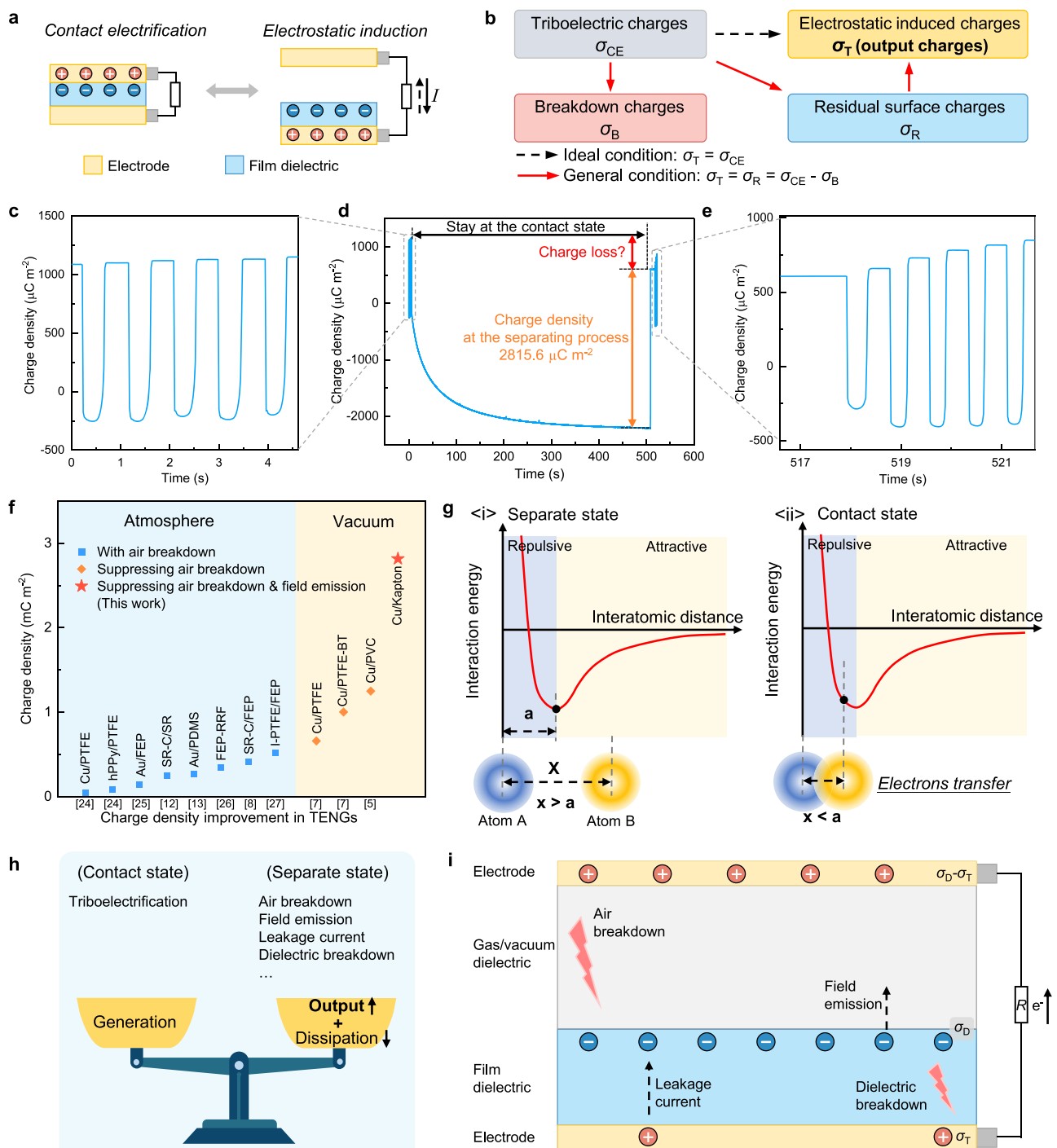

**Fig. 1 | Hypothesis of field emission in triboelectric nanogenerator. a** Working mechanism of TENGs based on contact electrification and electrostatic induction. **b** Schematic diagram shows the charges flowing process in a TENG. In ideal condition, the output charges of TENGs is equal to the triboelectric charges if there are no charge dissipations. In general conditions, partial triboelectric charges are often released by breakdown effects, and then the output charges are equal to the residual surface charges. **c**–**e** The output charge density of polyimide (6 μm) and copper under vacuum conditions. **c** The cyclic output charge density at 1 Hz; **d** the charge density at the separating process after contacting several minutes; **e** the cyclic output charge density at 1 Hz again. **f** The represented charge density in

contact electrification tested by TENGs[5,7,8,12,13,24–27]. **g** Electron transfer at the contact state. At the separate state, the distance between atom A and atom B is $x$, which is larger than the distance $a$ where two atoms are at equilibrium position. When $x < a$, the electronic clouds of two atoms are strongly overlapped, and electrons transfer occurs at this state. **h** Balanced charge density in TENGs considering both charge generation in the contact state and charge dissipations in the separate state. **i** Possible charge dissipations in TENGs, including air breakdown, field emission, leakage current, and dielectric breakdown. The surface charge density (SCD) on dielectric material is denoted as $\sigma_D$; the transferred charge density in the external circuit is denoted as $\sigma_T$. Source data are provided as a Source data file.

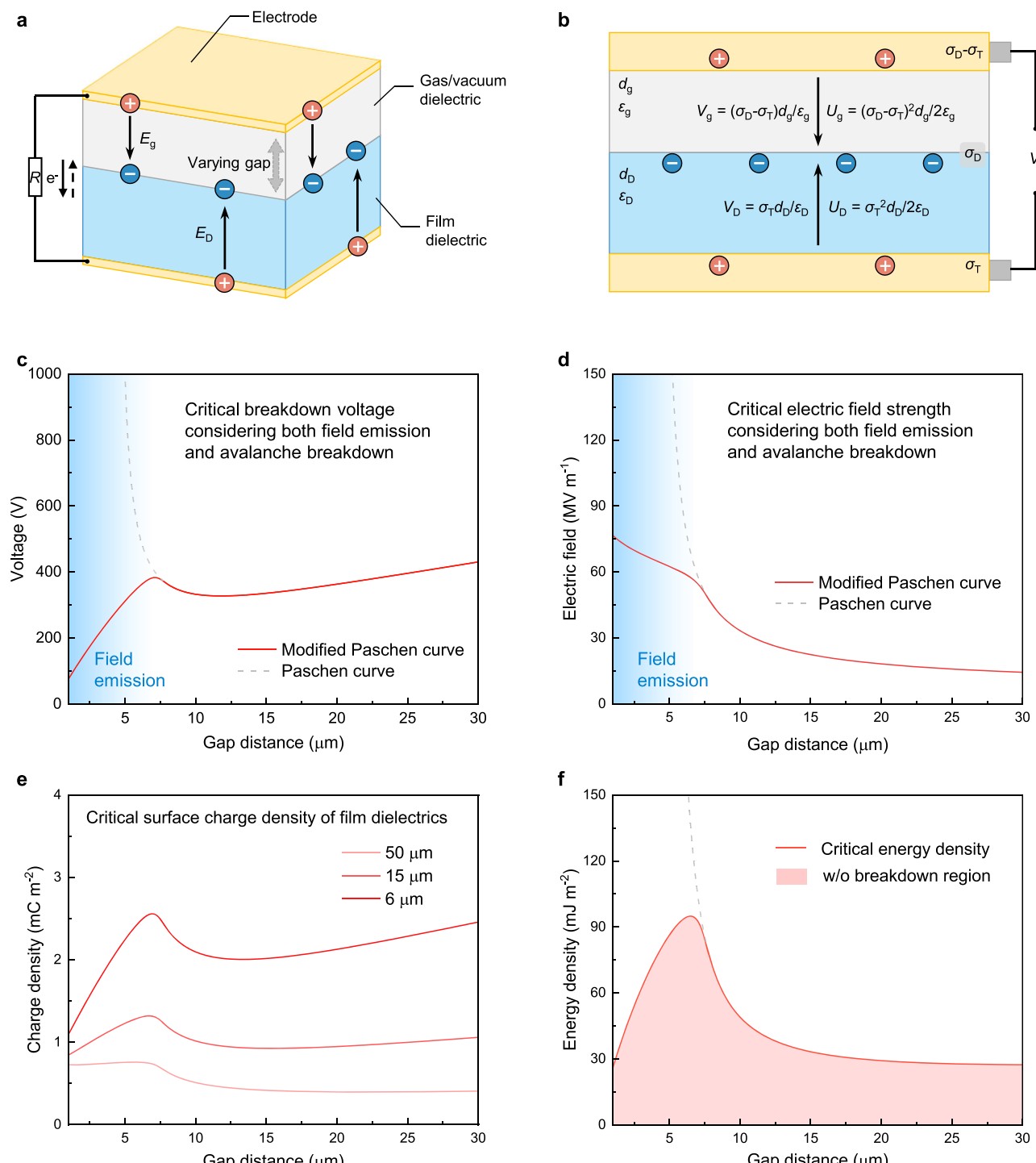

**Fig. 2 | Critical breakdown electric field for field emission in TENGs. a** Electric field distribution across the gas/vacuum dielectric ($E_g$) and the film dielectric ($E_D$) in TENGs with the varying gap. The arrow indicates the electric field. **b** Charge dynamics, and resulting voltage and energy changes. Voltage across the gas/vacuum dielectric ($V_g$) and the film dielectric ($V_D$) in TENGs with SCD of $\sigma_D$ and the transferred charge density of $\sigma_T$; energy density stored in the gas/vacuum dielectric ($U_g$) and the film dielectric ($U_D$). For the gas/vacuum dielectric layer, its thickness and permittivity are $d_g$ and $\varepsilon_g$, respectively; for the film dielectric layer, its thickness and permittivity are $d_D$ and $\varepsilon_D$, respectively. The arrow indicates the electric potential. **c** The relationship between critical breakdown voltage and gap distance. **d** The relationship between critical electric field and gap distance. The solid and dash lines are obtained by the ratio of voltage to gap distance from the modified Paschen curve and the Paschen curve, respectively. **e** The critical surface charge density of film dielectrics with various thicknesses. **f** The critical energy density in the gap w/ breakdown. The dash line represents the results described by the Paschen curve. Source data are provided as a Source data file.

related to contact electrification is more complex. From the perspective of breakdown theory, the possible breakdown effects are Townsend avalanche and field emission (Fig. 2c, d and Supplementary Note 1). Generally, the required electric field strength for field emission is higher than that for Townsend avalanche[28]. This is also accordance with the description in Eq. (2) that $E_g$ decreases with the increasing of $\sigma_T$, because of the gradual increased transferred charges in the external circuit as $d_g$ increases.

Under atmosphere conditions, Townsend avalanche and field emission can be intertwined, complicating their distinction, particularly at the micrometer scale[29]. The transferred charge density in TENGs can be described as the following equation[5]:

$$\sigma_T = \frac{\sigma_D d_g \varepsilon_D}{d_D \varepsilon_g + d_g \varepsilon_D} \tag{8}$$

By substituting $\sigma_T$ into Eq. (2), the critical SCD of film dielectric considering both Townsend avalanche and field emission in short-circuit condition can be described as follows.

$$\sigma_D = \frac{E_g\left(\varepsilon_g d_D + d_g \varepsilon_D\right)}{d_D} \tag{9}$$

It is worth noting that the critical electric field for field emission also relies on surface condition, materials properties, environmental factors, etc. Especially considering the unequally distributed surface charges at local states, the local electric field at these points as well as the edge of the surface could be higher. As shown in Fig. 2e, the critical SCD of film dielectrics varies with the thickness of the film dielectric. Correspondingly, the critical energy density in the gas capacitor is limited to the value indicated by the red line in Fig. 2f (calculated using Eq. (6) and the data in Fig. 2d).

Under vacuum conditions, however, these two phenomena can be decoupled in principle due to the avoidance of Townsend avalanche[30]. According to the previous works, the critical electric field ($E$) for field emission are assumed of $1 \times 10^8\,\text{V m}^{-1}$ to $5 \times 10^8\,\text{V m}^{-1}$, and the corresponding SCD ($\sigma$) for two charged materials is 0.885-to-4.425 mC m$^{-2}$ (Supplementary Fig. 3, according to the equation of $E = \sigma/\varepsilon_0$; $\varepsilon_0 = 8.85 \times 10^{-12}\,\text{F m}^{-1}$). It is noted that this is the open-circuit condition where no charge transfer in the external circuit. For the short-circuit condition, considering the transferred charges in the external circuit during gradual separation, the permitted $\sigma$ is higher than the above value, which also relies on the dielectric thickness and the relative permittivity.

For TENGs, the charge density has reached to the order of mC m$^{-2}$ under vacuum conditions. These values are close to the predicted threshold for field emission. Nevertheless, this raises fundamental questions: Is there field emission, and does field emission occur prior to dielectric breakdown in TENGs? Addressing these questions is crucial for understanding the upper limit of charge density in contact electrification and optimizing the performance of TENGs (Supplementary Note 2 and Supplementary Figs. 3–5).

## Effects of field emission on output characteristics of CS-TENG from contact to separate state

Given that the inherent capacitance of CS−TENG at the contact and separate states are different (Supplementary Fig. 6), the output performance of CS−TENG is discussed from two aspects: from contact to separate and from separate to contact. Particularly, a diode is parallel connected with the CS−TENG to achieve charge reset[31,32], leading to an even higher gap voltage and more pronounced field emission. More importantly, this design can avoid charge loss during the separating or contact processes affecting the experimental results mutually. Additionally, parasitic capacitance[31,33] should be carefully controlled before experiments (Supplementary Figs. 7, 8 and Supplementary Note 3). Here, all of the experiments are carried out under vacuum conditions to avoid the occurrence of air breakdown.

The effect of field emission on output characteristics of CS−TENG from contact to separate is studied firstly. With the upper electrode connected to the negative terminal of diode while the lower electrode connected to the positive terminal of diode, the diode is reverse biased and current flows through the load in the separation process; during the approaching process, the diode is forward biased and current flows

through the diode to realize charge reset in CS-TENG (Fig. 3a). Obviously, as the external load resistance increases, the duration of charge transfer in the external circuit prolongs, exhibiting typical characteristics of the RC circuit, and the peak output current gradually diminishes (Fig. 3b–d). It is noteworthy that an increase in load resistance not only prolongs the charge transfer time in the external circuit, but also has an impact on the gap voltage in TENGs[31], thereby influencing field emission. Experimental results reveal a gradual decrease in output charge with increasing load resistance, underscoring the influence of field emission on output performance (Fig. 3e). The corresponding output energy curve also corroborates this result. Consequently, the output voltage stabilized at around 10 kV with the resistance increasing (Fig. 3f), implying the limitation from field emission (Supplementary Note 4). Moreover, this data in Fig. 3f indicate that the maximum output energy of CS−TENG cannot be obtained directly due to the surface charge loss at the large external load (Supplementary Note 5).

An indirect method is then introduced to characterize the output energy cycle of CS−TENG by charging different capacitors (Fig. 3g and Supplementary Note 6)[23]. From the perspective of electric circuit, the CS−TENG in separate condition is $C_D$ and $C_g$ connected in series. If CS−TENG is connected with an external fixed capacitor ($C_f$), charges will distribute in these three capacitors according to the Kirchhoff's voltage law. Several groups of output charge and voltage can be obtained by changing $C_f$, and the output energy cycle of CS−TENG can be obtained if these groups of output charge and voltage are connected by a straight line in a voltage-charge figure ($V$−$Q$ figure), as shown in Fig. 3h. At a large $C_f$, the output charge is close to $Q_{SC}$. At a small $C_f$, the output voltage is close to $V_{OC}$. Specifically, field emission could exist with a small $C_f$, as the voltage in a TENG increases with a small $C_f$, leading to the collapse of $V$−$Q$ curve (the red line in Fig. 3i). Experimentally, when $C_f$ is only tens of picofarads, the voltage over time curve indicates charge loss during the separation process (Fig. 3j and k), implying the limitation from field emission again (Supplementary Note 7). By calculating the output energy of CS−TENG under different capacitors, our results reveal that at high voltages, the output energy curve collapses, deviating from the ideal output energy curve. Moreover, by varying the thickness of film dielectric, we observe little change in the output energy of CS−TENG, indicating that the output energy from contact to separate is independent of the thickness of film dielectric (Fig. 3l). Here, the maximum output energy density of CS−TENG at the separate state is -5.68 J m$^{-2}$ after regulating the field emission effect (As shown in Fig. 3h, the maximum stored energy in $C_f$ is a quarter of the maximum energy enclosed by the $V$−$Q$ curve when the output charge is a half of $Q_{SC}$.).

## Effects of field emission on output characteristics of CS-TENG from separate to contact state

We then investigated the effect of field emission on output characteristics of CS−TENG from separate to contact. With the upper electrode connected to the positive terminal of diode but the lower electrode connected to the negative terminal of diode, the diode is forward biased and current flows through the diode to realize charge reset in CS−TENG; during the approaching process, the diode is reverse biased and current flows through the load in the separation process (Fig. 4a). The output charge and energy density of CS−TENG from separate to contact state are also decreased at the large external load (Fig. 4b–d), indicating the field emission phenomenon again.

The indirect method to characterize the output energy cycle of CS−TENG by charging different capacitors is also adopted. The equivalent electric component of CS−TENG at the contact state is $C_d$ (Fig. 4e). As shown in Fig. 4f and Supplementary Fig. 9, the charge quantity charged into $C_f$ gradually decreases as $C_f$ decreases. Correspondingly, the output energy curve of CS−TENG in the contact state is plotted in Fig. 4g. It can be observed that under high voltage values,

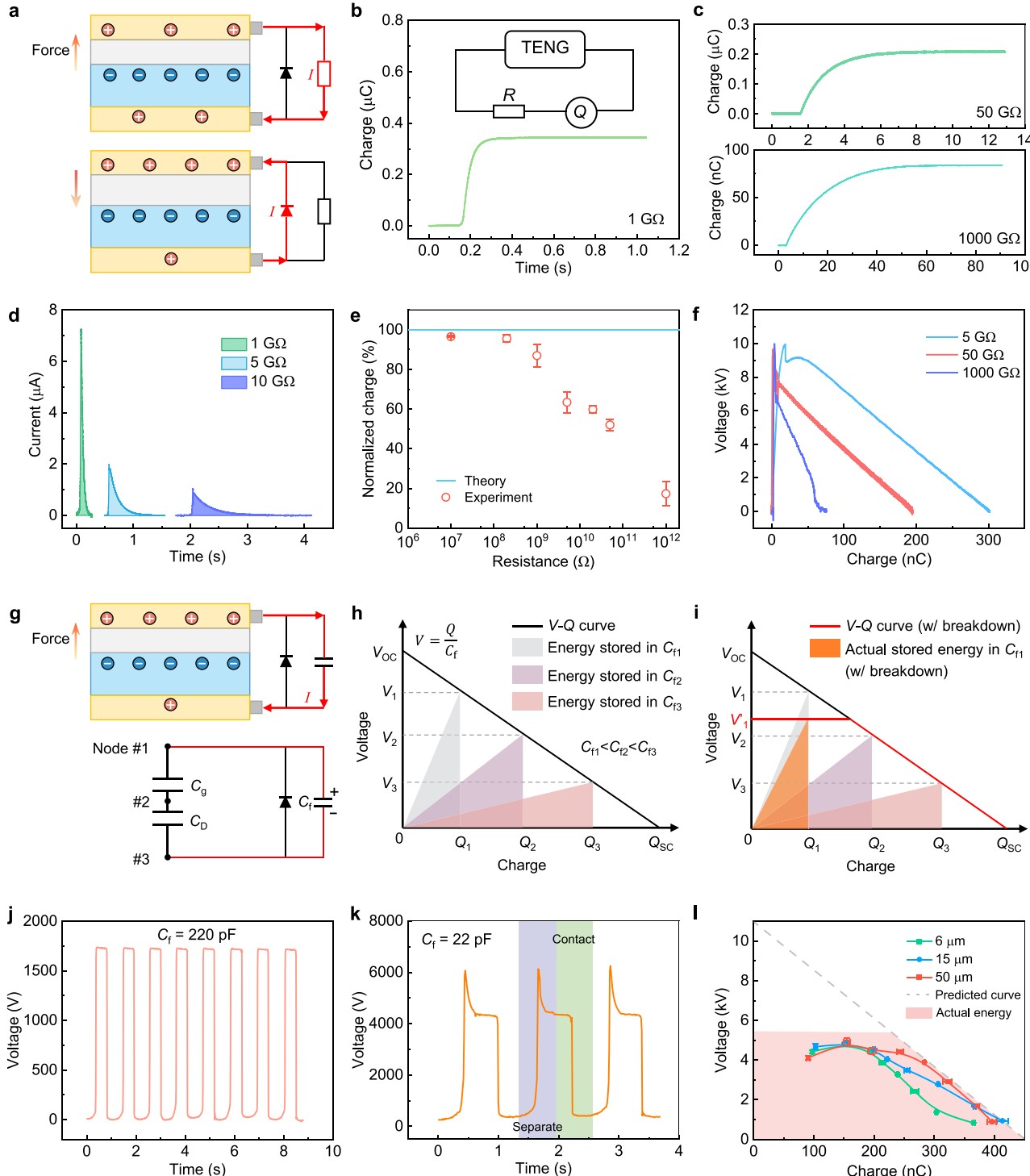

**Fig. 3 | Effects of field emission on output characteristics of CS-TENG from contact to separate state. a** Test circuit to maximize the output energy of CS-TENG on the load. A diode is parallel connected with the CS-TENG to achieve charge reset. **b, c** Output charge of CS-TENG with different load resistance. **d** Output current of CS-TENG with different load resistance. **e** Normalized charge of CS-TENG with different load resistances. Theoretically, it is predicted that surface charges keep stable at different load resistances. Data are presented as mean values ± SD, $n = 3$ independent experiments. **f** V–Q curves of CS-TENG with different load resistance. **g** Electric circuit of CS-TENG charging a capacitor and the corresponding equivalent circuit model. From the perspective of electric circuit, the CS-TENG in separate condition is a dielectric capacitor ($C_D$) and a gas capacitor ($C_g$) connected in series. $C_f$ is an external fixed capacitor. **h** Schematic diagram

shows that the V–Q curve of CS-TENG for charging different capacitors. A series of charge ($Q_1$, $Q_1$ and $Q_3$) and voltage ($V_1$, $V_1$ and $V_3$) can be obtained by changing the external fixed capacitor ($C_f$). By linearly connecting these charge and voltage points in the V–Q figure, the V–Q curve of CS-TENG can be obtained theoretically. **i** The actual obtained V–Q curve for charging different capacitors w/ breakdown. The breakdown effect limits the achievable maximum output voltage. The output voltage curve of CS-TENG for charging the capacitor of **j** 220 pF and **k** 22 pF. **l** Experimental V–Q curves of CS-TENG with different dielectric thickness. The gray dashed line shows the possible predicted V–Q curve w/o breakdown. Data are presented as mean values ± SD, $n = 5$ independent measurements. Source data are provided as a Source data file.

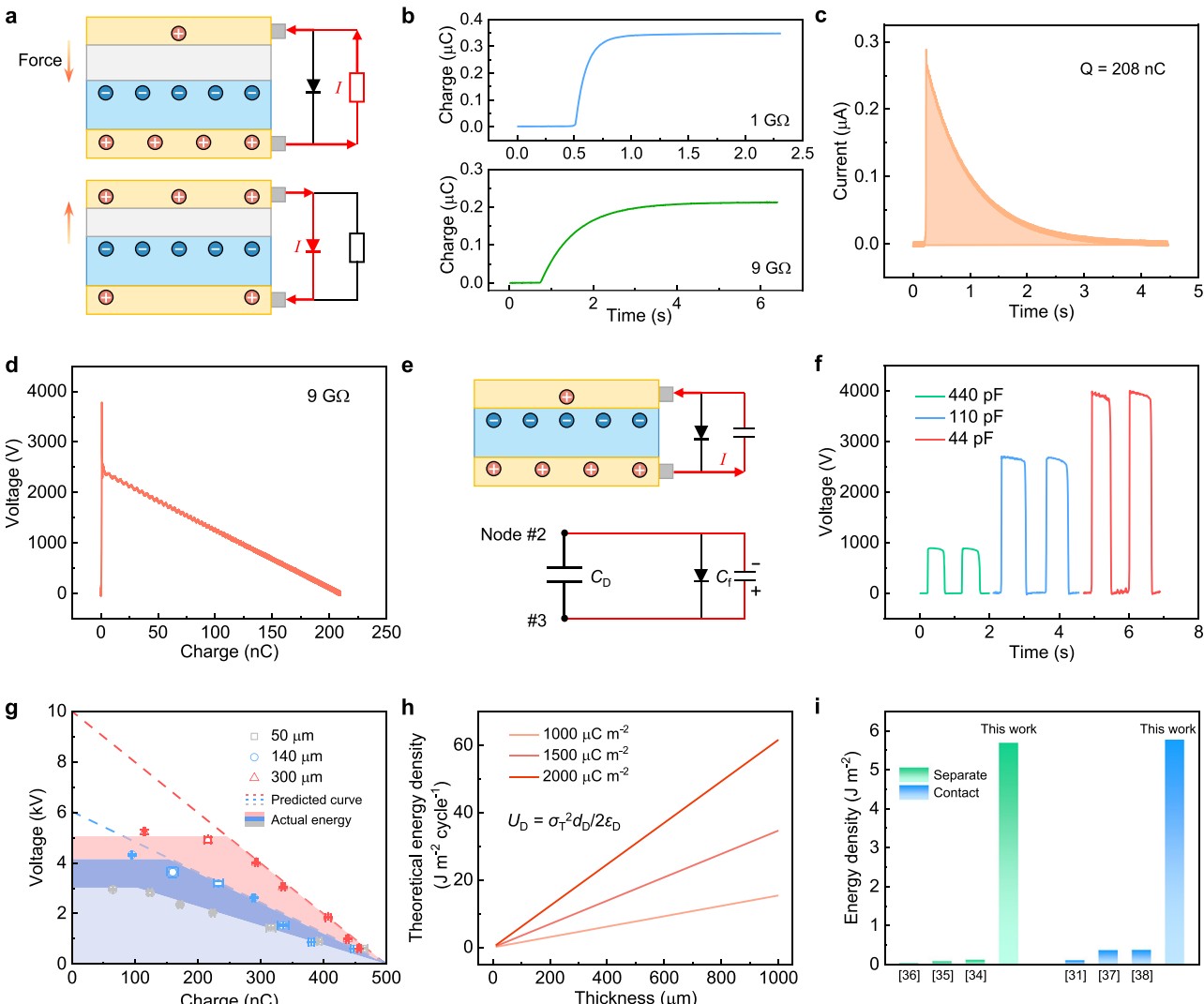

**Fig. 4 | Effects of field emission on output characteristics of CS−TENG from separate to contact state. a** Test circuit to maximize the output energy of CS−TENG on the load. A diode is parallel connected with the CS−TENG to achieve charge reset. **b** Output charge of CS−TENG with different load resistance. **c** Output current of CS−TENG at 9 GΩ. **d** V−Q curve of CS−TENG at 9 GΩ. **e** Electric circuit of CS−TENG charging a capacitor at the contact process and the corresponding equivalent circuit model. From the perspective of electric circuit, the CS−TENG in contact condition is a dielectric capacitor ($C_D$). $C_f$ is an external fixed capacitor. **f** The output voltage curve of CS−TENG for charging different capacitors at separate process. **g** V−Q curves of CS−TENG with different dielectric thickness. These curves are obtained by charging different capacitors with CS−TENG at contact process. The breakdown effect limits the achievable maximum output voltage. The dashed lines show the possible predicted V−Q curve w/o breakdown. Data are presented as mean values ± SD, n = 5 independent measurements. **h** Theoretical energy density of CS−TENG at the contact sate with different SCD and dielectric thickness. **i** The represented output energy density of CS−TENG comparing with the previous works[31,34−38]. Source data are provided as a Source data file.

the maximum output energy curve of CS−TENG in the contact state still showed varying degrees of collapse. Especially, as the thickness of film dielectric increases, the collapse of energy curve of CS−TENG becomes more severe. Because, as the thickness of film dielectric increases (resulting in a decrease in dielectric capacitance), the voltage applied to CS−TENG also increases, thereby intensifying the field emission effect. For a film dielectric of 300 μm, the maximum output energy density is 5.76 J m⁻² after regulating the field emission effect, which is much higher than previous reports and close to the theoretical maximum output energy density of 7.2 J m⁻² (Fig. 4h, i and Supplementary Note 8)[31,34−38]. It is noted that the output energy density of the TENG over an entire working cycle (from contact to separate and from separate to contact) reaches up to 11.44 J m⁻², highlighting the promising potential of TENGs for various applications like energy harvesting, sensing, contact-electro-catalysis, and beyond.

## Universality of the field emission effect

For the CS−TENG with different dielectric thicknesses, the degree of breakdown caused by field emission varies. Under conditions of equal SCD and gap distance ($\sigma_{D1} = \sigma_{D2}$, $d_{g1} = d_{g2}$) but different thicknesses of film dielectric ($d_{D1} < d_{D2}$) (Fig. 5a), a thicker film dielectric results in less charge transfer in the external circuit ($\sigma_{T1} > \sigma_{T2}$). This leads to a higher electric field, voltage, and energy density in the gap ($E_{g1} < E_{g2}$, $V_{g1} < V_{g2}$, $U_{g1} < U_{g2}$). Consequently, field emission is more likely to occur in the CS−TENG with the thicker film dielectric due to the stronger electric field. The calculated results also indicate a higher electric field value in the gap when the dielectric film is thicker (Fig. 5b). In the demonstrated devices, the measured SCD of CS−TENG with the film dielectric of 1090 μm indeed has the lowest value (Fig. 5c, different thicknesses of the film dielectric are achieved by layer upon layer to maintain a consistent surface as much as possible.).

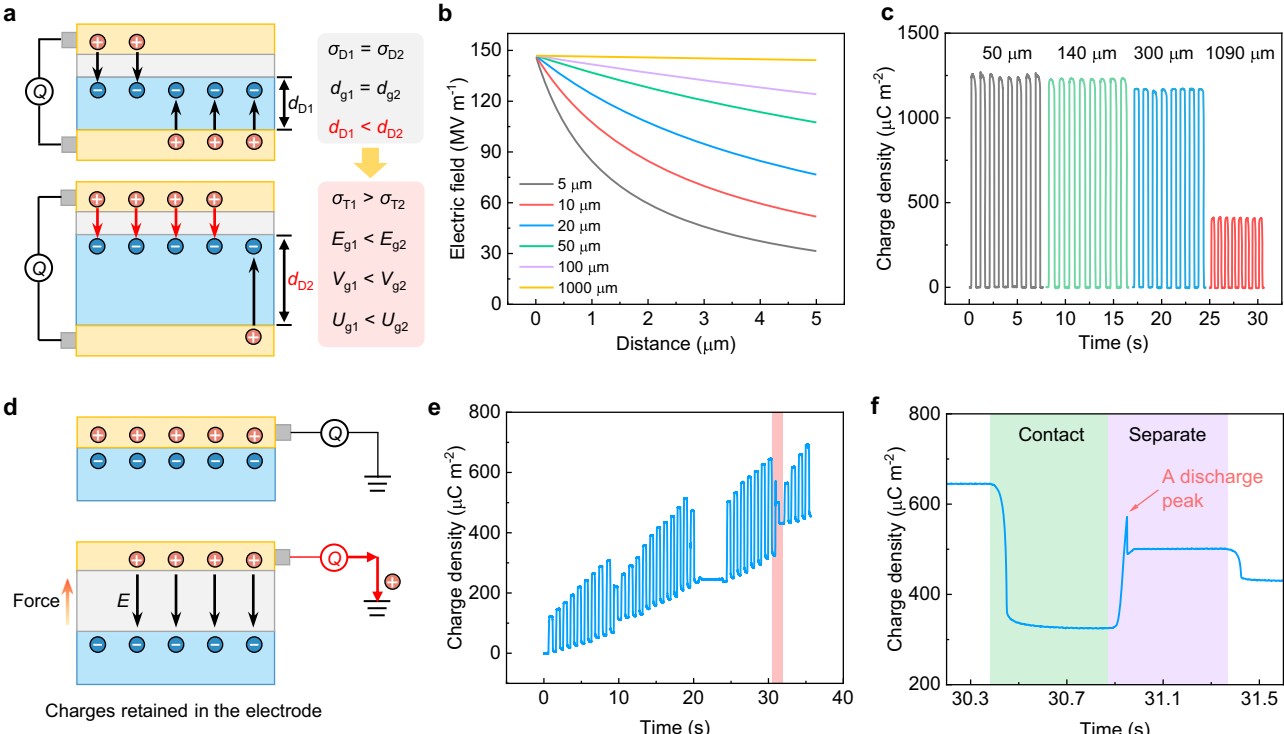

**Fig. 5 | Universality of the field emission effect in TENGs. a** Schematic diagram shows the short-circuit transferred charge of CS–TENG with the different dielectric thickness. The larger dielectric thickness can enhance the electric field across the gap, leading to enhanced field emission. **b** Calculated electric field in the gap of CS-TENG with the different dielectric thickness. The surface charge density of 1300 μC m⁻² is used for calculation. **c** Measured output charge of CS–TENG in short-circuit condition. **d** Test circuit of transferred charge in a S–TENG. An electrometer is directly connected with the electrode to monitor the charge transfer process. **e** Short-circuit transferred charge of S–TENG under vacuum conditions. The red shading area shows a representative discharge peak. **f** The enlarged figure shows a discharge peak in the separating process of the S–TENG at figure **e**. Source data are provided as a Source data file.

We also demonstrate the existence of field emission in the single-electrode mode TENG (S–TENG). Due to insufficient time for charge transfer in the external circuit, a strong electric field is present in the air gap during the separation process, especially in sub-micron separation gaps. The charge barely transfers in the external circuit, causing the electric field to be confined entirely within the gap between the two triboelectric layers (Fig. 5d). We tested $Q_{SC}$ of the S–TENG in vacuum, and the results showed that its output charge suddenly decays at around 300–400 μC m⁻² (Here, the maximum separated distance is 10 mm in the S–TENG.). According to theoretical calculations, the transferred charge at a separation distance of 10 mm is only about 20% of the surface charges (Supplementary Fig. 10), indicating that SCD on the triboelectric layer under this condition is about 1500–2000 μC m⁻². The electric field in the gap could reach the threshold for field emission, leading to the decay of surface charges. An obvious discharge peak during separation provides a solid evidence to support this point (Fig. 5e, f and Supplementary Fig. 11). Additionally, field emission also could occur in other sliding working mode TENGs, which should be carefully considered in different structures (Supplementary Fig. 12 and Supplementary Note 9).

**Limiting factors in TENGs**

It is noted that the output charge density of TENGs depends on the SCD of dielectric layer. Given that the simultaneously existed charge generation process and charge dissipation process, SCD should be the steady charge density, balanced the charging and discharging processes. Contact electrification is the only charging process, which occurs at the contact state where two atoms electronic cloud is strong overlapped. For the discharging process, it can be divided into two types: discharging into the outside and discharging into the

dielectric material. The represented phenomena of discharging into the outside are air breakdown and field emission. Air breakdown is the common discharging phenomenon in air condition, and field emission is often overlooked. The represented phenomena of discharging into the dielectric material are leakage current and dielectric breakdown. The leakage current has been demonstrated in many recent works, while the dielectric breakdown has nearly reported because of the high breakdown electric field for dielectric materials. Moreover, the dielectric breakdown is more likely to be occurred at the farthest separated state where the electric field across the dielectric material is the highest (Supplementary Fig. 13).

For TENGs with different structures, the electric field distribution is different, leading to the different limiting factor, which should be carefully investigated in actual conditions. More importantly, most of previous works focus on the dynamic charge transfer process under the short-circuit condition, often neglecting load conditions. However, it is crucial to recognize that SCD under the short-circuit condition often cannot be fully utilized under large load conditions, making the high SCD under the short-circuit condition is meaningless from a power generation perspective[23]. Therefore, understanding the dynamic charge transfer process under load conditions is vital for enhancing the output voltage and energy of TENGs.

It has been demonstrated in this work that the output charge, voltage, and energy are often limited by field emission, rather than dielectric breakdown, if the air breakdown effect is removed. In other words, field emission is the next bottleneck of TENGs. These results indicate that the SCD of TENGs, not only in short-circuit conditions but also in load conditions, is simultaneously decided by contact electrification, dielectric breakdown, field emission, and air breakdown,

described by the following equation:

$$\sigma_{\text{TENG}} = (\sigma_{\text{contact electrification}}, \sigma_{\text{dielectric breakdown}}, \sigma_{\text{field emission}}, \sigma_{\text{air breakdown}})\,\text{min} \tag{10}$$

where ($\sigma_{\text{contact electrification}}$) represents the charge generation process due to contact electrification; $\sigma_{\text{air breakdown}}$ and $\sigma_{\text{dielectric breakdown}}$ represent the discharge threshold from air breakdown and dielectric breakdown, respectively; $\sigma_{\text{field emission}}$ represents the discharge threshold from field emission. Previous studies concentrate on improving $\sigma_{\text{contact electrification}}$ and $\sigma_{\text{air breakdown}}$, and the SCD has approached to the threshold for field emission, so $\sigma_{\text{field emission}}$ should be paid more attention for further enhancing the performance of TENG in the future.

## Discussion

In summary, the well-suited design of CS–TENG combining with the charge-reset diode, and S-TENG, indicate the existence of field emission in TENGs, and we also demonstrate its significant impact on TENGs' electric properties, including the output charge, current, voltage, and energy. The surface charge loss from field emission in each TENG depends on the charge dynamic process that is attributed to SCD, electric field distribution, and external load conditions, etc., and the amount of surface charge loss would be random in each breakdown, so that practically the survival gap voltage should be smaller than the theoretical breakdown threshold rather than exactly equal. These findings challenge the previous understanding that triboelectric charge density and energy density of TENGs are primarily restricted by contact electrification, air breakdown, and dielectric breakdown effects. Instead, field emission, as a new limiting factor, should be considered to optimize the output performance of TENGs.

Our results have demonstrated that field emission occurs before dielectric breakdown; the present SCD under atmosphere conditions is close to the field emission threshold (Field emission is often coupled with Townsend avalanche called ion-enhanced field emission.), so $\sigma_{\text{field emission}}$ should be paid more attention for further enhancing the performance of TENGs in the future. It is important to note that the parameters related to field emission were obtained from metal-to-metal pairs in previous works, and there may be discrepancies when applying these parameters to dielectric materials[39–41]. Those parameters may require further refinement through ongoing research. Specifically, for charge excitation TENGs, the limitations of maximum effective charge density also could be field emission (Supplementary Fig. 14 and Supplementary Note 10). The potential method for restricting field emission would be introducing another external reverse electric filed to reduce the gap voltage, reducing the thickness (or increasing the permittivity) of film dielectric to decrease the interfacial electric field, and insulating the edge or corner of charged materials to reduce the local intensified electric field, etc[23].

The high charge density of 2.816 mC m$^{-2}$ achieved through contact electrification promotes also remind us to reconsider the upper limit of charge quantity generated by contact electrification. When two charged materials undergo contact electrification, their separation from zero distance to some fine distance inevitably triggers both field emission and air breakdown effects sequentially. Notably, field emission at micro scales has a higher electric field threshold than air breakdown, meaning that air breakdown will still occur even after field emission. Therefore, the upper limit of charge quantity generated by contact electrification could be higher than what we have measured if discharge processes are avoided. This underscores the importance of paying closer attention to charge dissipation processes in future research. Moreover, the substantial charge density achieved through contact electrification may offer new insights and opportunities across various scientific communities and application fields, such as electrostatic adsorption and separation, contact-electro-catalysis, energy harvesting, sensors, and beyond.

## Methods
### Fabrication of the TENG
The CS–TENG was fabricated by the following steps. A circular polytetrafluoroethylene plate was cut by using a laser cutter (PLS6.75, Universal Laser System) with the diameter of 30 mm and the thickness of 3 mm, which is used as the substrate. A copper foil (diameter: 30 mm; thickness: 0.05 mm) was cut and adhered to the surface of the substrate as the electrode layer. A circular acrylic plate (diameter: 22.6 mm; thickness: 3 mm) was cut as the other substrate. A copper foil (diameter: 22.6 mm; thickness: 0.05 mm) was adhered to the surface of the substrate as another electrode layer, and a polyimide film with an area of 4 cm$^2$ was adhered to this electrode layer as the triboelectric layer. For the S–TENG in Fig. 5 and the CS–TENG in Fig. 1, the device area is 1 cm$^2$.

### COMSOL simulation
The parameters in the S–TENG for simulation are as follows: the surface charge density of film dielectric is 400 μC m$^{-2}$, the relative dielectric constant of film dielectric is 3.5, the air gap increases from 0.5 mm to 10 mm.

### Electrical performance characterization
Experiments were carried out under a home-made high vacuum system (the ultimate vacuum of the molecular pump is about $5 \times 10^{-5}$ Pa) to realize a stable environment. The working temperature is around 298 K. The output charge of TENGs (CS–TENG and S–TENG) was measured by an electrometer (Keithley 6514); the output voltage was measured by an electrometer (Trek 370) or monitored by monitoring the voltage of the charging capacitor; the output current was measured by an electrometer (Keithley 6514). Data acquisition was performed using a NI-6218, and real-time data acquisition and analysis were achieved using LabVIEW.

## Data availability
All relevant data supporting the key findings of this study are available within the article and its Supplementary Information files. Source data are provided with this paper.

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

## Acknowledgements

Research was supported by the National Key R & D Project from Minister of Science and Technology (2021YFA1201602) and National Natural Science Foundation of China (Grant No. U21A20147, 62204017 and 22109013).

## Author contributions

D.L., Y.G., and J.W. conceived the idea. D.L. and Y.G. designed the experiments and performed data measurements. W.Q., L.Z., L.H., C.Y., B.J., and B.Z. helped with the experiments. D. L. drafted the manuscript. Z. L.W. and J.W. revised the manuscript and supervised this work. All the authors discussed the results and commented on the manuscript.

## Competing interests

The authors declare no competing interests.
