## [Transparent Peer Review file · Nature Communications]

Field emission effect in triboelectric nanogenerators

Corresponding Author: Professor Jie Wang

Version 0:

Reviewer comments:

Reviewer #1

(Remarks to the Author)

This manuscript presents a significant advancement in the field of triboelectric nanogenerators (TENGs) by identifying and thoroughly investigating the field emission effect as a newly discovered limiting factor in the performance of TENGs. The authors demonstrate that field emission occurs before dielectric breakdown, introducing a critical factor that influences the maximum achievable charge density and energy output in TENG systems. The study provides a well-structured analysis supported by both theoretical modeling and experimental validation, including a clear explanation of the mechanisms involved and extensive characterization of charge density and output performance using CS-TENG and S-TENG setups. The methodology is rigorous, with comprehensive data presentation and effective use of supplementary figures to reinforce key findings. The results significantly contribute to the understanding of contact electrification dynamics and provide valuable insights for enhancing TENG performance, making this work a substantial contribution to the field of energy harvesting technologies. Therefore, I suggest that this manuscript can be accepted in "Nature Communication" after the following some revisions.

Comment 1: "This manuscript provides comprehensive performance data for the TENG devices but lacks sufficient details on the reproducibility of the results and their statistical reliability. If data from repeated experiments are available, including statistical analysis such as standard deviation or error bars on key performance graphs would improve the scientific rigor and clarity of the findings."

Comment 2: "The interaction between field emission and air breakdown can vary depending on environmental factors such as temperature and humidity during experiments. Additional clarification on these conditions would provide a more comprehensive understanding of the charge dissipation mechanisms and their impact on TENG performance."

Reviewer #2

(Remarks to the Author)

This is a very interesting work, which reveals a new important factor that affects the outcome of TENG. The manuscript is well written. All the data and figures can support the argument very well. I think the manuscript can be accepted after addressing the following points.

1, In the introduction part, the authors talked about using "interface lubrication" to enhance the performance. The authors cited the work done by Zhou L et al. As long as I know, that is not the first work that initiates the strategy of using interface lubrication. Even the authors didn't say which one is the first work. But I strongly believe we should respect the pioneer on the work, especially when we plan to publish work on Journals like Nature Communication.

2, The authors claim that the field emission effect exists in all working modes. But what I can see is that the manuscript only reported the contact-separation model together with single electrode mode. How about the other working modes, e.g., free standing, sliding, TVNG, DC-TENG?

3, This work focuses on the material of PI, does this field emission effect also apply to other materials?

4, This work only did the tests under vacuum conditions, what will happen if we run the experiment under normal open air condition? What will happen if we increase the temperature of humidity?

Version 1:

Reviewer comments:

Reviewer #1

(Remarks to the Author)

The authors provided the appropriate response and additional results in response to referees; as a result, the revised manuscript was improved by thorough revision process. The reviewer thinks that the manuscript is acceptable to this journal without any change.

Reviewer #2

(Remarks to the Author)

I am pretty happy with this revised version. I think all my previous comments or questions have been well addressed now. Thus, I think it can be accepted for publication now.

Point-by-point responses to the reviewers' comments

We sincerely thank the reviewers for carefully review our work, which are indeed very helpful to make the paper more solid and smooth. We have revised our manuscript very carefully. The following responses have been prepared to address all of the reviewers' comments in a point-by-point fashion. (**Comments in black, responses in Blue, revisions in red**)

REVIEWER COMMENTS

Reviewer #1 (Remarks to the Author):

This manuscript presents a significant advancement in the field of triboelectric nanogenerators (TENGs) by identifying and thoroughly investigating the field emission effect as a newly discovered limiting factor in the performance of TENGs. The authors demonstrate that field emission occurs before dielectric breakdown, introducing a critical factor that influences the maximum achievable charge density and energy output in TENG systems. The study provides a well-structured analysis supported by both theoretical modeling and experimental validation, including a clear explanation of the mechanisms involved and extensive characterization of charge density and output performance using CS-TENG and S-TENG setups. The methodology is rigorous, with comprehensive data presentation and effective use of supplementary figures to reinforce key findings. The results significantly contribute to the understanding of contact electrification dynamics and provide valuable insights for enhancing TENG performance, making this work a substantial contribution to the field of energy harvesting technologies. Therefore, I suggest that this manuscript can be accepted in "Nature Communication" after the following some revisions.

Response: We highly appreciate the reviewer for carefully reviewing our work, and thank your generous comments on our research work.

Comment 1: "This manuscript provides comprehensive performance data for the TENG devices but lacks sufficient details on the reproducibility of the results and their statistical reliability. If data from repeated experiments are available, including

statistical analysis such as standard deviation or error bars on key performance graphs would improve the scientific rigor and clarity of the findings.”

Response: Thank you very much for your careful review and professional comment, which is indeed very helpful to make the manuscript more solid. We have revised the data with repeated experiments to ensure the reproducibility and reliability of the TENG performance.

We have added other two independent experiments as **Figure R1** (also named **Supplementary Fig. 1**) to support the results in **Figs. 1c-e**. As shown in **Figure R1**, the charge dissipation still exists even when the contact time is extended to surpass an hour.

The added **Figure R1** is as follows:

Figure R1 (also named **Supplementary Fig. 1**). The output charge density of polyimide (6 μm) and copper under vacuum conditions. **a** and **b** Output charge density with different contact time. The charge dissipation still exists even when the contact time is extended to surpass an hour.

We have added error bars in **Fig. 3e**, **Fig. 3l**, and **Fig. 4g**. The revised figures are shown below.

Figure R2 (also named **Fig. 3e**). **Normalized charge of CS-TENG with different load resistances.** The error bars correspond to data from three independent measurement.

Figure R3 (also named **Fig. 3f**). **Experimental V-Q curves of CS-TENG with different dielectric thickness.** The error bars correspond to data from five independent measurement.

Figure R4 (also named **Fig. 4g**). **V-Q curves of CS-TENG with different dielectric thickness.** The error bars correspond to data from three independent measurement.

We have added **Supplementary Fig. 11** to demonstrate the universality of field emission in TENGs with other dielectric materials.

Figure R5 (also named **Supplementary Fig. 11**). **Short-circuit transferred charge of S-TENG of different materials under vacuum conditions. a PVC. b** The enlarged figure at figure a. **c FEP. d** The enlarged figure at figure c.

We have added **Figure R6** (also named **Supplementary Fig. 12** and described as **Supplementary Note 9**) to demonstrate the universality of field emission in sliding mode TENGs. For the sliding mode TENGs, including free-standing mode TENG, lateral sliding mode TENG, and direct-current TENG (DC-TENG), the most likely location

for breakdown is the corner between the slider and the stator for these working modes because of the intensified electric field, as we described in the previous paper (*Nature Communications*, 2024, 15, 4167). To clearly illustrate the field emission effect in sliding mode TENGs, we take the DC-TENG as an example. The working mechanism of DC-TENG is based on triboelectrification and electrostatic breakdown in atmosphere condition. In principle, the DC-TENG should have no output in vacuum condition where the air breakdown effect is fully removed, as the description in our previous work. However, we find that a large external force, higher than the previous work, can produce more triboelectric charges on the surface of dielectric layer, so the intensified electric field around the corner of the FE in DC-TENG can breakdown the gap even in vacuum conditions. Different from the utilization of air breakdown between the charge collection electrode (CCE) and the triboelectric layer (TL) in atmosphere conditions (① in **Supplementary Fig. 12a** and **b**), DC-TENG utilizes field emission between the frictional electrode (FE) and TL in vacuum conditions (② in **Supplementary Fig. 12a** and **c**). This breakdown will result in the fast current flow in external circuit with high peak values but short-duration pulse (**Supplementary Fig. 12d** and **e**), which is different from the constant current output of DC-TENG in atmospheric conditions. Therefore, the working mechanism of DC-TENG in vacuum condition is based on triboelectrification and electrostatic breakdown that occurred at around FE, rather than the previous breakdown domain around CCE. The COMSOL simulation result also suggests the possibility of occurrence of field emission, leading to charge release from FE to TL (**Supplementary Fig. 12c**). It is noted that field emission around the corner of FE could happen only when surface charge density is very high.

Figure R6 (also named **Supplementary Fig. 12**). **Field emission in sliding mode TENG.** **a** Two breakdown domains in DC-TENG. ① is the breakdown phenomenon between CCE and TL; ② is the breakdown phenomenon between FE and TL. **b** Working mechanism of DC-TENG in atmosphere condition arising from air breakdown. **c** Working mechanism of DC-TENG in vacuum condition arising from field emission. The enlarged figure shows the schematic electric field intensified at the corner of FE. **d** Output charge curves of DC-TENG in atmosphere and vacuum conditions. The enlarged figure shows the separated steps of charge growth in vacuum condition. **e** Output current curve of DC-TENG in vacuum condition. The enlarged figure shows the separated current peak of DC-TENG. **f** The simulated electric field distribution in a DC-TENG. The electric field at the corner around the left of FE is easy beyond 10^9 V m^{-1} . The parameters in the DC-TENG for simulation are as follows: the relative dielectric constant of film dielectric is 2.5; the surface charge density of film dielectric is $500 \mu\text{C m}^{-2}$.

The revised part in the present manuscript is as follows:

We have added “An obvious discharge peak during separation provides a solid evidence to support this point (**Fig. 5e, f and Supplementary Fig. 11**). Additionally, field emission also could occur in other sliding working mode TENGs, which should

be carefully considered in different structures (**Supplementary Fig. 12** and **Supplementary Note 9**)." in the part of "Universality of the field emission effect".

Supplementary Note 9. Field emission effect in sliding mode TENGs.

For the sliding mode TENGs, including free-standing mode TENG, lateral sliding mode TENG, and direct-current TENG (DC-TENG), the most likely location for breakdown is the corner between the slider and the stator for these working modes because of the intensified electric field.⁶ To clearly illustrate the field emission effect in sliding mode TENGs, we take the DC-TENG as an example. The working mechanism of DC-TENG is based on triboelectrification and electrostatic breakdown in atmosphere condition. In principle, the DC-TENG should have no output in vacuum condition where the air breakdown effect is fully removed, as the description in our previous work. However, we find that a large external force, higher than the previous work, can produce more triboelectric charges on the surface of dielectric layer, so the intensified electric field around the corner of the FE in DC-TENG can breakdown the gap even in vacuum conditions. Different from the utilization of air breakdown between the charge collection electrode (CCE) and the triboelectric layer (TL) in atmosphere conditions (① in **Supplementary Fig. 12a** and **b**), DC-TENG utilizes field emission between the frictional electrode (FE) and TL in vacuum conditions (② in **Supplementary Fig. 12a** and **c**). This breakdown will result in the fast current flow in external circuit with high peak values but short-duration pulse (**Supplementary Fig. 12d** and **e**), which is different from the constant current output of DC-TENG in atmospheric conditions. Therefore, the working mechanism of DC-TENG in vacuum condition is based on triboelectrification and electrostatic breakdown that occurred at around FE, rather than the previous breakdown domain around CCE. The COMSOL simulation result also suggests the possibility of occurrence of field emission, leading to charge release from FE to TL (**Supplementary Fig. 12c**). It is noted that field emission around the corner of FE could happen only when surface charge density is very high.

Comment 2: “The interaction between field emission and air breakdown can vary depending on environmental factors such as temperature and humidity during experiments. Additional clarification on these conditions would provide a more comprehensive understanding of the charge dissipation mechanisms and their impact on TENG performance.”

Response: Thank you very much for your careful review and professional comments. We completely agree with you that the interaction between field emission and air breakdown can vary depending on environmental factors such as temperature and humidity during experiments.

From the fundamental principles of field emission and air breakdown, both of them are indeed highly influenced by the environmental factors such as temperature and humidity. Generally, the electric field strength for air breakdown is significantly lower than the critical value needed for field emission (approximately three orders of magnitude difference). This is why air breakdown is often a focus of research under atmosphere conditions. In contrast, field emission is typically studied under very high-pressure conditions or in high vacuum conditions to completely decouple it from air breakdown. In our manuscript, we also did the experiments under vacuum conditions (the ultimate vacuum of the molecular pump is about 5×10^{-5} Pa) at the temperature of around 298 K to separate the two phenomena: field emission and air breakdown. This allowed us to investigate the effect of field emission on the performance of TENGs.

Under atmosphere conditions, field emission is often coupled with Townsend avalanche called ion-enhanced field emission in some special conditions. Therefore, the critical charge density for field emission should be paid more attention for further enhancing the performance of TENGs in the future. It is important to note that the parameters related to field emission were obtained from metal-to-metal pairs in previous works, and there may be discrepancies when applying these parameters to dielectric materials. Those parameters may require further refinement through ongoing research, as discussed in the manuscript. Additionally, regarding the effects of temperature and humidity on the performance of TENGs, we have provided a detailed discussion in our previous paper (*Nature Communications*, 2022, 13, 6019); even in

vacuum conditions, although there is no air breakdown effect, surface charges also could be dissipated by thermionic emission at high temperatures, thus coupling with the field emission effect.

Therefore, different charge dissipation phenomena may often occur simultaneously, and it is impossible to study the impact of environmental factors separately. We think that designing well-controlled experiments to isolate the individual types of dissipation is crucial for thoroughly understanding the dissipation mechanisms as we stated in this paper. This will enable us to fully comprehend the interaction between charge dissipation mechanisms. We look forward to improved experimental designs to investigate these two phenomena in atmosphere conditions.

The revised part in the present manuscript is as follows:

We have added “Experiments were carried out under a home-made high vacuum system (the ultimate vacuum of the molecular pump is about 5×10^{-5} Pa) to realize a stable environment. **The working temperature is around 298 K.**” in **Methods** section.

Reviewer #2 (Remarks to the Author):

This is a very interesting work, which reveals a new important factor that affects the outcome of TENG. The manuscript is well written. All the data and figures can support the argument very well. I think the manuscript can be accepted after addressing the following points.

Response: We highly appreciate the reviewer for carefully reviewing our work, and thank your generous comments on our research work.

1, In the introduction part, the authors talked about using “interface lubrication” to enhance the performance. The authors cited the work done by Zhou L et al. As long as I know, that is not the first work that initiates the strategy of using interface lubrication. Even the authors didn’t say which one is the first work. But I strongly believe we should respect the pioneer on the work, especially when we plan to publish work on Journals like Nature Communication.

Response: Thank you very much for pointing out this problem. We have cited the paper that initiates the strategy of using interface lubrication in TENGs.

The revised part in the present manuscript is as follows:

We have added “Suppressing air breakdown, including the use of ultrathin dielectric materials^{8,9}, interface lubrication^{10,11}, and structural design^{11,12}, can significantly increase the surface charge density, even by orders of magnitudes.⁶” in the **Introduction** section.

We have added “

10. Wu, J., Xi, Y. & Shi, Y. Toward wear-resistive, highly durable and high performance triboelectric nanogenerator through interface liquid lubrication. *Nano Energy* **72**, 104659 (2020).” in the **References** section.

2, The authors claim that the field emission effect exists in all working modes. But what I can see is that the manuscript only reported the contact-separation model together with single electrode mode. How about the other working modes, e.g., free standing, sliding,

TVNG, DC-TENG?

Response: Thank you very much for your careful review and professional comment. We also provide a detailed analysis of field emission effect in other working modes TENGs.

For the sliding mode TENGs, including free-standing mode TENG, lateral sliding mode TENG, and direct-current TENG (DC-TENG), the most likely location for breakdown is the corner between the slider and the stator for these working modes because of the intensified electric field, as we described in the previous paper (*Nature Communications*, 2024, 15, 4167). To clearly illustrate the field emission effect in sliding mode TENGs, we take the DC-TENG as an example. The working mechanism of DC-TENG is based on triboelectrification and electrostatic breakdown in atmosphere condition. In principle, the DC-TENG should have no output in vacuum condition where the air breakdown effect is fully removed, as the description in our previous work. However, we find that a large external force, higher than the previous work, can produce more triboelectric charges on the surface of dielectric layer, so the intensified electric field around the corner of the FE in DC-TENG can breakdown the gap even in vacuum conditions. Different from the utilization of air breakdown between the charge collection electrode (CCE) and the triboelectric layer (TL) in atmosphere conditions (① in **Supplementary Fig. 12a** and **b**), DC-TENG utilizes field emission between the frictional electrode (FE) and TL in vacuum conditions (② in **Supplementary Fig. 12a** and **c**). This breakdown will result in the fast current flow in external circuit with high peak values but short-duration pulse (**Supplementary Fig. 12d** and **e**), which is different from the constant current output of DC-TENG in atmospheric conditions. Therefore, the working mechanism of DC-TENG in vacuum condition is based on triboelectrification and electrostatic breakdown that occurred at around FE, rather than the previous breakdown domain around CCE. The COMSOL simulation result also suggests the possibility of occurrence of field emission, leading to charge release from FE to TL (**Supplementary Fig. 12c**). It is noted that field emission around the corner of FE could happen only when surface charge density is very high.

Figure R7 (also named **Supplementary Fig. 12**). **a** Two breakdown domains in DC-TENG. ① is the breakdown phenomenon between CCE and TL; ② is the breakdown phenomenon between FE and TL. **b** Working mechanism of DC-TENG in atmosphere condition arising from air breakdown. **c** Working mechanism of DC-TENG in vacuum condition arising from field emission. The enlarged figure shows the schematic electric field intensified at the corner of FE. **d** Output charge curves of DC-TENG in atmosphere and vacuum conditions. The enlarged figure shows the separated steps of charge growth in vacuum condition. **e** Output current curve of DC-TENG in vacuum condition. The enlarged figure shows the separated current peak of DC-TENG. **f** The simulated electric field distribution in a DC-TENG. The electric field at the corner around the left of FE is easy beyond 10^9 V m^{-1} . The parameters in the DC-TENG for simulation are as follows: the relative dielectric constant of film dielectric is 2.5; the surface charge density of film dielectric is $500 \mu\text{C m}^{-2}$.

For the TVNG, its working mechanism is based on tribovoltaic effect, which is different from the conventional TENGs based on triboelectrification and electrostatic induction/breakdown. Generally, the voltage in TVNG is much smaller than the voltage in conventional TENGs. Therefore, a much high charge density could be required to cause field emission in TVNG, which still needs to be exploited in the future.

The revised part in the present manuscript is as follows:

We have added this part as **Supplementary Fig. 12** and **Supplementary Note 9**.

Supplementary Note 9. Field emission effect in sliding mode TENGs.

For the sliding mode TENGs, including free-standing mode TENG, lateral sliding mode TENG, and direct-current TENG (DC-TENG), the most likely location for breakdown is the corner between the slider and the stator for these working modes because of the intensified electric field.⁶ To clearly illustrate the field emission effect in sliding mode TENGs, we take the DC-TENG as an example. The working mechanism of DC-TENG is based on triboelectrification and electrostatic breakdown in atmosphere condition. In principle, the DC-TENG should have no output in vacuum condition where the air breakdown effect is fully removed, as the description in our previous work. However, we find that a large external force, higher than the previous work, can produce more triboelectric charges on the surface of dielectric layer, so the intensified electric field around the corner of the FE in DC-TENG can breakdown the gap even in vacuum conditions. Different from the utilization of air breakdown between the charge collection electrode (CCE) and the triboelectric layer (TL) in atmosphere conditions (① in **Supplementary Fig. 12a** and **b**), DC-TENG utilizes field emission between the frictional electrode (FE) and TL in vacuum conditions (② in **Supplementary Fig. 12a** and **c**). This breakdown will result in the fast current flow in external circuit with high peak values but short-duration pulse (**Supplementary Fig. 12d** and **e**), which is different from the constant current output of DC-TENG in atmospheric conditions. Therefore, the working mechanism of DC-TENG in vacuum condition is based on triboelectrification and electrostatic breakdown that occurred at around FE, rather than the previous breakdown domain around CCE. The COMSOL simulation result also suggests the possibility of occurrence of field emission, leading to charge release from FE to TL (**Supplementary Fig. 12c**). It is noted that field emission around the corner of FE could happen only when surface charge density is very high.

3, This work focuses on the material of PI, does this field emission effect also apply to other materials?

Response: Thank you very much for your careful review and professional comment. From the mechanism described in our paper, field emission effect is universal for various dielectric materials. We also provide another two materials including polyvinyl chloride (PVC) and fluorinated ethylene propylene (FEP) to demonstrate this, by utilizing the single-electrode TENG.

Figure R8 (also named **Supplementary Fig. 11**). Short-circuit transferred charge of S-TENG of different materials under vacuum conditions. **a** PVC. **b** The enlarged figure at figure a. **c** FEP. **d** The enlarged figure at figure c.

The revised part in the present manuscript is as follows:

We have added this part as **Supplementary Fig. 11**.

4, This work only did the tests under vacuum conditions, what will happen if we run the experiment under normal open air condition? What will happen if we increase the temperature of humidity?

Response: Thank you very much for your careful review and professional comment. Generally, the electric field strength for air breakdown is significantly lower than the

critical value needed for field emission (approximately three orders of magnitude difference), so air breakdown is very easy to occur than field emission under atmosphere conditions, leading to surface charges dissipated into air. Particularly in open-circuit condition (a special case for load conditions), the maximized surface charge density is restricted to only around $50 \mu\text{C m}^{-2}$. This has been carefully discussed in our previous work (*Nature Communications*, 2022, 13, 6019).

Then, if the ultrathin dielectric layer is used for restricting the air breakdown effect in TENGs, the surface charge density of dielectric layer can be improved to a higher value (*Advanced Materials*, 2014, 26, 6720-6728; *Applied Materials Today*, 2020, 18, 100496). As we described in **Fig. 1h**, surface charges are in dynamic equilibrium between charge generation, charge dissipation, and charge output. Even in a well-controlled experiment under atmosphere conditions, the possibility of charge dissipations from adsorption, leakage charge, etc., increases due to the slow charge accumulation process, resulting in a limited surface charge density which couldn't cause field emission.

With the assistance of charge pump technology to overcome the slow charge accumulation process and to break the limits of triboelectrification on surface charge density (*Nano Energy*, 2018, 49, 625-633; *Nature Communications*, 2018, 9, 3773; *Nature Communications*, 2019, 10, 1426), a higher charge density in TENGs can be achieved. As the effective charge density in charge pump TENGs increases, field emission could be considered as another limitation factor. This has been carefully discussed in **Supplementary Note 10**.

Overall, under atmosphere conditions, other charge dissipations like adsorption, charge leakage, air breakdown, etc., rather than field emission often dominate the charge dissipation process in TENGs. However, for those TENGs with high surface charge density, field emission could be another limitation factor restricting the maximized surface charge density, which should be carefully considered depending on the specific working mode, structure, atmosphere conditions, and load conditions.

Additionally, regarding the effects of temperature and humidity on the performance of TENGs, we have provided a detailed discussion in our previous paper (*Nature*

Communications, 2022, 13, 6019); even in vacuum conditions, although there is no air breakdown effect, surface charges also could be dissipated by thermionic emission at high temperatures, thus coupling with the field emission effect.

Therefore, different charge dissipation phenomena may often occur simultaneously, and it is impossible to study the impact of environmental factors separately. We think that designing well-controlled experiments to isolate the individual types of dissipation is crucial for thoroughly understanding the dissipation mechanisms as we stated in this paper. This will enable us to fully comprehend the interaction between charge dissipation mechanisms. We look forward to improved experimental designs to investigate these two phenomena in atmosphere conditions.

Point-by-point responses to the reviewers' comments

We sincerely thank the reviewers for carefully review our work, which are indeed very helpful to make the paper more solid and smooth. We have revised our manuscript very carefully. The following responses have been prepared to address all of the reviewers' comments in a point-by-point fashion. (**Comments in black, responses in Blue**)

REVIEWER COMMENTS

Reviewer #1 (Remarks to the Author):

The authors provided the appropriate response and additional results in response to referees; as a result, the revised manuscript was improved by thorough revision process. The reviewer thinks that the manuscript is acceptable to this journal without any change.

Response: We highly appreciate the reviewer for carefully reviewing our work, and thank your positive comments on our research work and recommendation for publications.

Reviewer #2 (Remarks to the Author):

I am pretty happy with this revised version. I think all my previous comments or questions have been well addressed now. Thus, I think it can be accepted for publication now.

Response: We highly appreciate the reviewer for carefully reviewing our work, and thank your positive comments on our research work and recommendation for publications.